# AI-Based Facial Emotion Analysis in Infants During Complimentary Feeding: A Descriptive Study of Maternal and Infant Influences

**DOI:** 10.3390/nu17193182

**Published:** 2025-10-09

**Authors:** Murat Gülşen, Beril Aydın, Güliz Gürer, Sıddika Songül Yalçın

**Affiliations:** 1Autism, Special Mental Needs and Rare Diseases Department, Turkish Ministry of Health, Ankara 06800, Turkey; 2Department of Pediatrics, School of Medicine, Başkent University, Ankara 06490, Turkey; 3Department of Pediatrics, Balıkesir Atatürk City Hospital, Balıkesir 10100, Turkey; 4Department of Pediatrics, School of Medicine, Hacettepe University, Ankara 06230, Turkey; ssyalcin22@gmail.com

**Keywords:** Artificial Intelligence, facial expression analysis, OpenFace, complementary feeding, infants, breastfeeding

## Abstract

**Background/Objectives**: Infant emotional responses during complementary feeding offer key insights into early developmental processes and feeding behaviors. AI-driven facial emotion analysis presents a novel, objective method to quantify these subtle expressions, potentially informing interventions in early childhood nutrition. We aimed to investigate how maternal and infant traits influence infants’ emotional responses during complementary feeding using an automated facial analysis tool. **Methods**: This multi-center study involved 117 typically developing infants (6–11 months) and their mothers. Standardized feeding sessions were recorded, and OpenFace software quantified six emotions (surprise, sadness, fear, happiness, anger, disgust). Data were normalized and analyzed via Generalized Estimating Equations to identify associations with maternal BMI, education, work status, and infant age, sex, and complementary feeding initiation. **Results**: Emotional responses did not differ significantly across five food groups. Infants of mothers with BMI > 30 kg/m^2^ showed greater surprise, while those whose mothers were well-educated and not working displayed more happiness. Older infants and those introduced to complementary feeding before six months exhibited higher levels of anger. Parental or infant food selectivity did not significantly affect responses. **Conclusions**: The findings indicate that maternal and infant demographic factors exert a more pronounced influence on infant emotional responses during complementary feeding than the type of food provided. These results highlight the importance of integrating broader psychosocial variables into early feeding practices and underscore the potential utility of AI-driven facial emotion analysis in advancing research on infant development.

## 1. Introduction

Emotions play a central role in human communication, conveyed through nonverbal cues such as body language, gestures, vocal tones, and facial expressions. From infancy, these cues are vital for forming social bonds and facilitating interactions. For example, social smiles are fundamental to mother–infant bonding, and mothers exhibit specific brain responses to their child’s distress [1]. Although emotions may be intuitively understood, studying the mechanisms behind emotion recognition in an objective, scientific manner remains a formidable challenge.

In 1978, Paul Ekman and his team established the Facial Action Coding System (FACS), defining 44 separate muscle movements called action units (AUs) [2]. While originally developed for adults, infant-adapted approaches such as the Baby FACS have since been introduced to account for age-related differences in facial structure and expression [3]. Nevertheless, the fundamental AU framework remains highly relevant and provides the basis for automated facial analysis. Each AU corresponds to specific facial muscle activities, such as AU6 (cheek raiser) and AU12 (lip corner puller). Examining these units on their own or in various combinations enables the interpretation of intricate emotional expressions, including happiness, sadness, and disgust. For example, AU6 and AU12 both have higher intensities in a happy face. The specific AUs used in this study are provided in Figure 1. Despite its utility, manually encoding these movements is time-consuming and susceptible to human error. Technological progress, however, has facilitated the emergence of automated facial analysis systems like OpenFace [4]. This open-source software utilizes deep neural networks to assess facial images or videos, calculating AU intensities with high accuracy and efficiency [5]. Compared to manual methods, OpenFace accelerates the process by analyzing frames rapidly and impartially, thus reducing potential biases and saving considerable time.

Breastfeeding is the primary nutritional source for the first six months of life. After this period, complementary foods are introduced to bolster continued growth and development. This period is critical for establishing healthy eating habits and preferences, with long-term implications for a child’s well-being [6,7]. A well-structured introduction to complementary foods paves the way for a diverse diet, while poor nutritional balance during this critical window may hamper a child’s growth and development [8,9,10]. Identifying feeding challenges at an early stage is therefore vital to ensure adequate nutrition and avert persistent challenges [11]. Feeding difficulties in young children are a widespread public health concern, impacting about 25% of typically developing children [12]. Because the initial signs can be vague, these issues can often be overlooked, leading to delayed diagnosis and intervention.

The interaction between the infant, caregiver, and food during the complementary feeding period can be analyzed through the Theory of Planned Behavior (TPB), Social Cognitive Theory (SCT), Social Norms Theory, the normative components of the Theory of Reasoned Action, and the Transtheoretical Model (TTM) [13]. The Theory of Planned Behavior (TPB) highlights the role of perceived behavioral control and self-efficacy in determining whether caregivers adhere to complementary feeding recommendations. A caregiver’s confidence in feeding practices may shape both the choice of foods offered and the infant’s subsequent responses. From the perspective of Social Cognitive Theory (SCT), caregivers often model the feeding practices of their peers and family members, influencing decisions such as the introduction of certain textures or foods at an earlier stage, which in turn may elicit distinct emotional and facial expressions in infants. Infants’ facial expressions in response to food are shaped by observational learning and prior feeding experiences, which influence their expectations and reactions to new foods. Social Norms Theory and the Theory of Reasoned Action further explain how family dietary habits and cultural norms shape infants’ acceptance or rejection of foods. Social Norms Theory specifically accounts for strong cultural or familial pressures that may encourage early introduction of solid foods, even when such practices conflict with current health recommendations. The Transtheoretical Model (TTM) describes the process of behavioral adaptation to new foods as occurring in distinct stages. Initially, in the contemplation stage, infants may display expressions of surprise or disgust when encountering an unfamiliar food. However, as they undergo repeated exposure (preparation and action stages), their acceptance may increase, leading to more positive facial expressions and, ultimately, incorporation of the food into their routine diet (maintenance stage). Despite the relevance of these theoretical frameworks, limited research has explored the facial expressions of infants aged 6–11 months during their initial exposure to new foods, highlighting a gap in the literature that warrants further investigation.

Facial expressions offer valuable insights into emotional responses, and infants exhibit distinct facial reactions to different tastes [14,15]. However, there is limited research on the use of automated facial emotion analysis during the complementary feeding period, with most studies either focusing on manual observation or broader infant emotion recognition outside this context [16,17,18].

We hypothesize that maternal and infant characteristics (such as maternal BMI, education, working status, and the infant’s age and sex) will influence the type and intensity of emotional expressions displayed by infants during complementary feeding, as measured by AI-based facial analysis. Our previous work utilized the same software to examine infants’ emotional reactions to different complementary foods [19]. This study aims to build upon this foundation by using a distinct methodology that focuses on child characteristics. By examining specific facial expressions, our goal is to identify factors influencing feeding behavior and social and emotional responses in diverse settings. Ultimately, this research seeks to facilitate earlier detection of feeding problems, guide targeted interventions, and support improved nutritional outcomes for infants.

## 2. Materials and Methods

### 2.1. Study Protocol

This descriptive, multi-center study took place at the Pediatric Department of Başkent University Ankara Hospital and at Balıkesir Atatürk State Hospital. The investigation targeted infants between six and eleven months of age who were typically developing and under maternal care during routine pediatric follow-up visits. Infants were excluded if they had any congenital anomalies, were born preterm (<37 weeks), had metabolic conditions requiring specialized feeding (e.g., phenylketonuria), were part of multiple births (twins/triplets), or had experienced an acute illness within the preceding week.

Upon obtaining informed consent, mother–infant pairs were enrolled in the study. Pregnancy history, birth weight, and breastfeeding status were obtained through an information form.

Sociodemographic characteristics (mother’s age, education level, employment status, birth order, child’s age, gender), feeding-related characteristics (birth weight, breastfeeding status, age of complementary feeding initiation, food preparation method, the person/people responsible for feeding the baby), and feeding behaviors (parental and child food selectivity) were recorded using a study form. For food selectivity, mothers were asked to report the foods that both parents and their infants refused to eat. For growth indicators, infant weight and length were measured during the scheduled healthcare visits. The mother’s weight and height were recorded, and the mother’s BMI was calculated. Although there is no universally accepted definition, we defined picky eaters as children who consume an insufficient variety of foods while rejecting substantial amounts of both familiar and unfamiliar foods [6,20].

Each mother was asked to select one item from five predefined food groups: meat-based dishes, dairy products, vegetable dishes, fruit purees, and grain-based foods with preference given to foods the infant had already tried at least once before the study (see Table A1). The food groups were designed based on Infant and Young Child Feeding (IYCF) guidelines and national nutritional guide [19,21,22]. All infants had been introduced to solids at home prior to enrollment.

The feeding sessions were self-recorded by the mothers at home and provided at subsequent appointments. Mothers were given standardized instructions on camera placement, lighting, and distance (approximately one meter from the infant) to maximize video quality. If a submitted video did not meet these criteria (e.g., major face occlusion or poor illumination), a repeat recording was requested. For each participant, the weight-for-age z-score (WAZ), length-for-age z-score (LAZ) were calculated using WHO Anthro software [23].

Ethical approval was granted by the Başkent University Ethics Committee (KA24/68-24/54, 28 February 2024).

### 2.2. Capturing and Processing the Video

Video data were gathered with a fixed-position camera positioned approximately one meter away from each infant in a sufficiently lit setting, maintaining at least 1920 × 1080-pixel resolution. Each of the five food categories was offered in sequence, allowing enough time for the child to chew and swallow before proceeding. A brief pause of around 10 s, along with a small amount of water, was provided between different food items to reset the infant’s sense of taste.

The recorded videos were then processed using OpenFace 2.0 on a Windows-based computer. Although the software can analyze 18 separate AUs, this study focused on AUs pertinent to emotional expressions (Figure 1). OpenFace computes a confidence score for every frame, which decreases in lower-quality situations, such as significant movements of the head or when the face is partially obstructed by an object or an extremity. Frames scoring below 90% confidence, constituting fewer than 1% of the total frames, were omitted from further examination to achieve higher reliability.

To enhance the precision of the emotion analysis, we employed a time-averaging filter that smooths the intensity values of AUs by averaging them over consecutive frames within each segment. After applying this method, we combined the intensity scores from two specific AUs associated with each of the six fundamental emotions to derive an aggregated measure of emotion intensity.

### 2.3. Statistics

Statistical analysis was performed using IBM Statistical Package for the Social Sciences (SPSS) version 23.0 (IBM Corp., Armonk, NY, USA). Categorical variables were given in percentile distribution. The distribution of continuous variables was assessed with the Kolmogorov–Smirnov test, histograms, kurtosis, and skewness measures. For normally distributed data, the mean and standard deviation (SD) were calculated. The percentile distribution of facial expressions for six different emotions was determined. The data on facial expressions were right-skewed. Therefore, an IDF-normal transformation, “a two-step approach for transforming non-normally distributed continuous variables to normal”, was applied [24]. To calculate the percentage of each IDF-corrected facial expression in the total, we used the following formula:$$\mathrm{percentage}\;\mathrm{of}\;x_{y}=\;\frac{x_{y}\times 100}{\sum_{y=1}^{6}x_{y}}$$

x: facial expression score

y1–6: Surprise, Sadness, Fear, Happiness, Anger, Disgust

Generalized Estimating Equations estimated differences in each “IDF-corrected emotional response” given during complementary feeding across different food groups and infant–mother pair factors (maternal age, maternal education, maternal working status, maternal BMI, birth order, infant age, infant’s sex, initiation age for complementary food, still breastfed, preparation of food for infant, infant’s WAZ). Overall, 117 subjects with 5 levels (within subject effect) were analyzed with identity link and normal distribution. Emotional responses to various food groups during complementary feeding in infants were also analyzed for associations with parental–infant food selectivity and infant age, using Generalized Estimating Equations.

Having an assistant to help with baby care and the mother’s working status were highly correlated; thus, only working status was included in the model. Estimated marginal means with 95% Wald confidence interval were calculated. Pairwise contrasts were performed with LSD method. A *p*-value of <0.05 was considered statistically significant.

## 3. Results

The study population consisted of 117 mother–infant pairs. General characteristics of the population are given in Table 1.

### 3.1. Population Characteristics

The mothers had a mean age of 31.0 ± 4.2 years, with 36.8% younger than 30 years, 46.2% between 30 and 34 years, and 17.1% aged 35 years or older. Most mothers (67.5%) had a university education or higher, and 32.5% were employed. The mean BMI of mothers was 24.5 ± 3.6, with 65.8% having a BMI below 25 km/m^2^. Assistance with baby care was reported by 33.3% of mothers.

Regarding eating behaviors, 10.3% of mothers and 15.4% of fathers were classified as picky eaters, with at least one picky eater parent present in 23.1% of families. Additionally, 28.2% of infants were identified as picky eaters.

The infants had a mean enrollment age of 8.8 ± 2.0 months, with 51.3% aged 6–8 months and 48.7% aged 9–11 months. The mean gestational duration was 38.7 ± 0.9 weeks, and the mean birth weight was 3243 ± 403 g. The sex distribution was nearly equal (48.7% male, 51.3% female). Complementary feeding was initiated at an average of 5.7 ± 0.5 months, with 43.6% starting at five months and 54.7% at six months. At the time of the study, 67.5% of infants were still being breastfed.

Regarding nutritional status, the mean weight-for-age Z score was 0.32 ± 0.88, with 6.0% of infants having a Z score below −1, 71.8% between −1 and +1, and 22.2% above +1. The majority (74.4%) were firstborn children. In terms of food preparation, mashing (75.2%) was more commonly used than grating (24.8%).

When asked about their favorite foods, approximately half of the infants (48.7%) were reported to favor yogurt, while 17.9% favored bananas. When asked about disliked foods, 21.4% of the infants were reported not to consume any food reluctantly. The most disliked food group was the meat group (32.5%), with specific items ranked as follows: 23.1% meat (13.7% in minced form, 9.4% in meatball form), 6% egg, 1.7% fish, and 1.7% chicken. However, food selectivity in children was found to be 28.2%. Although some infants were reported to dislike certain foods, they were still observed to consume them.

To better understand infants’ reactions to the most disliked food group, which is the meat group, a further analysis was conducted focusing on their facial expressions during consumption of meat (Table 2). A total of 44 infants were given minced meat, with 14 of them reported as disliking it. Additionally, 28 infants were given meatballs, and 10 of them were marked as disliking them. However, statistical analysis showed no significant differences in facial expressions between infants who were reported to dislike minced meat or meatballs and those who liked them (*p* > 0.05).

### 3.2. Emotions

A total of 585 facial emotional intensity measurements from 117 infants were analyzed. Percentile distribution for face expression intensities during complementary feeding given in Table 3. The three most frequently observed expressions were surprise (24.2%), sadness (20.0%), and anger (19.1%). The least observed expression was disgust, at 9%.

Although mean scores varied slightly, no statistically significant differences in any of the six measured emotions emerged across the five food groups (Figure 2). Despite the lack of within-subject effects by food type, multiple between-subject factors (maternal and infant characteristics, as well as food preparation practices) were significantly associated with specific emotional responses in infants (Table 4 and Figure 3).

Surprise: Infants of mothers with BMI > 30 km/m^2^ displayed significantly higher levels of surprise (26.3 [24.2–28.4]) than those of mothers with BMI <25 km/m^2^ (23.0 [21.4–24.6]) or BMI 25–29 km/m^2^ (22.8 [20.9–24.8]) (*p* = 0.007). Female infants (24.3 [22.8–25.9]) showed slightly higher levels of surprise than male infants (23.7 [22.1–25.3]) (*p* = 0.002). Infants presented with mashed foods (25.5 [24.2–26.8]) demonstrated higher levels of surprise compared to those offered grated foods (22.6 [20.7–24.4]) (*p* = 0.002)m.

Happiness: Infants of mothers with a university-level education or higher (7.8 [7.0–8.6]) showed more happiness compared to those whose mothers had less than a university education (6.3 [5.4–7.3]) (*p* = 0.008). Infants of non-working mothers (7.7 [7.0–8.5]) had higher happiness scores than working mothers (6.4 [5.4–7.4]) (*p* = 0.019).

Anger: Infants of working mothers displayed greater anger (20.7 [18.7–22.7]) than infants of non-working mothers (18.1 [16.3–19.9]) (*p* = 0.026). Older infants (9–11 months) exhibited marginally higher anger (20.4 [18.6–22.2]) compared to younger infants (6–8 months, 18.4 [16.6–20.2]) (*p* = 0.050). Infants introduced to complementary feeding before 6 months (20.7 [18.4–22.9]) demonstrated a higher anger response than those introduced at 6 months (18.2 [16.7–19.7]) (*p* = 0.034).

Disgust: Infants of non-working mothers (9.6 [8.8–10.4]) showed higher disgust scores than those of working mothers (8.4 [7.5–9.3]) (*p* = 0.025). Disgust responses differed by maternal BMI category, with infants of mothers in the 25–29 range (9.6 [8.7–10.5]) showing higher disgust than those with BMI <25 (8.3 [7.4–9.1]); (*p* = 0.050). Mothers with a BMI >30 did not differ significantly from the other groups.

Sadness, Fear, Other Factors: Across the measured demographic factors, no significant main effects emerged for sadness or fear (*p* > 0.05). While there were slight numerical differences (e.g., fear was marginally higher among older infants and working mothers), these did not reach statistical significance. Birth order, mother’s age, still breastfeeding status, and weight-for-age z-score (WAZ) did not show statistically significant associations with any of the six emotional responses (*p* > 0.05).

When associations with parental–infant food selectivity and infant age were analyzed with Generalized Estimating Equations, there were no significant differences in infants’ emotional responses across the five food groups. Generally, older infants (9–11 months) exhibited higher fear scores compared to younger infants (6–8 months). Neither the presence of infant selective eating nor parental selective eating was associated with significant differences in infants’ emotional responses during complementary feeding (Table 5).

## 4. Discussion

The findings from this AI-driven analysis of emotional responses during complementary feeding in infants provide valuable insights into how different demographic factors can influence infants’ emotions. Our study utilized the advanced capabilities of the OpenFace software to analyze facial expressions associated with key emotions, offering a detailed understanding of the nuanced emotional landscape during this critical developmental period.

### 4.1. Emotional Responses and Mother–Infant Characteristics

The association between higher maternal BMI and increased Surprise responses in infants may arise from variations in feeding practices or food selections linked to maternal obesity. Mothers with a higher BMI might offer a broader array of foods or employ distinct feeding techniques that provoke greater surprise, possibly due to novel flavors or unexpected textures. The slightly elevated Surprise in female infants compared to males could reflect a potential gender-based disparity in expressiveness or sensitivity to taste stimuli. Females may be more reactive to tastes, a trait that could reflect inherent differences in emotional processing or socialization patterns during feeding. The increased surprise response to mashed foods compared to grated ones suggests that infants may find the smoother texture more difficult to anticipate. This reaction could stem from their familiarity with more varied consistencies, making the uniform nature of mashed foods unexpected. Furthermore, the adhesive properties of mashed foods and their tendency to spread more rapidly in the oral cavity may further contribute to this heightened response.

The greater Happiness exhibited by infants of mothers with higher education levels might be attributed to more engaging or playful feeding interactions, fostering a positive mealtime atmosphere. Likewise, infants of non-working mothers showed higher Happiness possibly because non-working mothers may dedicate more time to creating enjoyable feeding experiences, or their consistent presence could provide infants with a sense of security that elevates happiness. Conversely, infants of working mothers might experience reduced caregiver interaction during mealtimes, potentially diminishing their positive emotional expressions.

Infants of working mothers displayed greater Anger responses. This could indicate frustration or a desire for attention during feeding, an interactive period where infants may expect engagement from their primary caregiver. The reduced availability of working mothers throughout the day might heighten infants’ emotional demands during these sessions, manifesting as increased Anger. Older infants also showed marginally higher Anger, potentially due to their advancing ability to express displeasure with certain foods more clearly. This developmental shift suggests that emotional expressiveness strengthens with age. In our study, approximately half of the infants had started complementary feeding before the 6th month. This finding is consistent with the current literature in our country [10,25]. Interestingly, infants introduced to complementary feeding before 6 months, which is prior to recommended age, exhibited higher levels of anger, potentially due to negative experiences associated with early feeding. These experiences may include gastrointestinal discomfort or exposure to unfamiliar tastes, leading to aversive reactions. Additionally, caregivers who initiate feeding prior to the recommended age may adopt stricter or less responsive feeding styles, potentially exacerbating infants’ anger by disregarding their cues.

Infants of non-working mothers demonstrated higher disgust scores. This might stem from non-working mothers having more time to prepare and try diverse foods, including those with stronger or less familiar flavors that may trigger Disgust. In contrast, working mothers may have to rely on a narrower range of convenient options, limiting infants’ exposure to distasteful stimuli.

In terms of sadness and fear, there were some slight numerical differences such as marginally higher Fear in older infants and those of working mothers, but these did not achieve statistical significance. This suggests that Sadness and Fear may be less prominently expressed during feeding or less influenced by the variables studied. The small sample size or the specific context of feeding sessions, which typically do not evoke intense distress, might also account for these non-significant findings.

The higher lactose content of human milk compared to the milk of other mammals may influence the infant’s taste preferences in complementary feeding [26]. In our study, birth order, maternal age, breastfeeding status, and WAZ showed no statistically significant associations with any emotional responses. These factors may exert minimal direct influence on infants’ emotional reactions during feeding, or their effects could be indirect, mediated through unexamined variables such as caregiver interaction quality or feeding frequency.

Interestingly, although some mothers reported that their infants disliked minced meat, they still offered it to them instead of selecting an alternative meat-based food, such as chicken or fish. Among the infants given minced meat, 14 out of 44 were reported as disliking it, while 10 out of 28 infants given meatballs were similarly reported as disliking them. However, no significant differences were observed in their facial expressions, suggesting that emotional responses may not reliably indicate food preferences at this stage of development. Therefore, facial expressions alone may not be a strong indicator of food acceptance or rejection in infants aged 6–11 months.

Furthermore, no significant association was found between parental food selectivity and infants’ facial expressions. In other words, whether or not parents themselves were selective eaters did not show a meaningful correlation with their infants’ emotional responses during feeding. Additionally, maternal reports of their infants’ food aversion did not align with actual food refusal, emphasizing the limitations of relying solely on parental perceptions to assess infant food preferences.

### 4.2. Implications for Early Childhood Feeding Practices

The significant associations between maternal and infant demographic factors and emotional responses underscore the importance of considering these variables when developing feeding strategies and interventions. For instance, recognizing that maternal education and employment status can influence infants’ emotions during feeding can inform the creation of targeted educational programs and support systems for mothers. Moreover, the link between maternal BMI and infant emotional responses highlights the potential benefits of maternal health interventions in improving both maternal and infant well-being. Also, having an additional caregiver may influence the child’s one-on-one interactions, potentially affecting their responses during the feeding period.

These findings also suggest that healthcare providers should adopt a more holistic approach when addressing feeding difficulties, considering not only the physical aspects of feeding but also the emotional and psychosocial context. Early identification of infants at risk of negative emotional responses during feeding could lead to more effective interventions, promoting better nutritional outcomes and healthier emotional development [27].

Importantly, feeding is not a one-directional process but a dynamic feedback loop. Parents make choices about what foods to offer, infants respond emotionally, and these reactions shape parents’ subsequent decisions about feeding. For instance, if a child consistently shows anger or disgust toward a certain food, a parent may avoid offering it in the future, while repeated expressions of happiness may reinforce continued provision of that food. Over time, these reciprocal interactions accumulate into recognizable feeding patterns and dietary preferences. Highlighting this cyclical nature of feeding underscores that infant emotions are not only outcomes of parental decisions but also active signals that guide future parental choices.

### 4.3. Relevance to Health Behavior Theories

According to the Theory of Planned Behavior, limited time and resources among working mothers may undermine opportunities for regular responsive interactions, possibly increasing infant anger and reducing happiness. Higher maternal education, on the other hand, may boost self-efficacy or clarify health goals, allowing for more consistent and positive feeding practices. In some communities, mothers feel they must begin solids around five months because, as they might say, “My baby needs solids now.” This pressure can manifest in greater frustration if the infant is not yet physiologically or emotionally ready, aligning with our finding of higher anger.

From the perspective of Social Cognitive Theory, parental reactions to specific foods can influence an infant’s approach to those foods. A key limitation of our study is that we did not simultaneously assess the facial expressions of the person feeding the infant, which may have played a role in shaping the infant’s response through observational learning.

Additionally, in line with Social Norms Theory and the Theory of Reasoned Action, infants’ acceptance or rejection of foods is shaped by familial and cultural dietary norms. Parents’ attitudes toward specific foods may impact both the infant’s facial expressions and their long-term food preferences. While we inquired about parental food selectivity, our findings indicated that certain foods were not even introduced to infants because they were not prepared at home, preventing further analysis. In societies with cultural food preferences and selective eating patterns, large-scale studies are needed to explore variations in infants’ and children’s facial expressions in response to food.

Furthermore, our study was conducted as a single-timepoint assessment, limiting our ability to observe changes in facial expressions over repeated exposures. The Transtheoretical Model recognizes that mothers may be at varying stages of readiness. Those still contemplating the best approach to complementary feeding can be more susceptible to time constraints, social influence, or uncertainty about whether early introduction is truly beneficial. Mothers who have moved into the action or maintenance stage may be more likely to offer consistent and evidence-based feeding approaches, which can help foster more positive emotions such as happiness or surprise in their infants. Repeated exposure to certain foods during the preparation and action stages can lead to increased acceptance, with positive facial expressions emerging over time and eventual dietary incorporation. However, further research is needed to explore how infants’ facial expressions evolve with repeated exposure to different food groups.

Future studies should employ longitudinal designs to examine changes over time and assess both the infant’s and the feeder’s facial expressions to better understand the interactive dynamics of complementary feeding. Integrating maternal beliefs, social norms, and self-efficacy into feeding interventions can help align recommendations with caregivers’ readiness, ultimately enhancing both the emotional quality of feeding and infant well-being.

### 4.4. Strengths and Limitations

A major strength of this study is the use of AI-driven facial analysis, which provides an objective and precise measure of infants’ emotional responses. This technology allows for the capture of subtle facial expressions that might be missed by human observers, offering a more comprehensive understanding of infant emotions. Additionally, it significantly shortens the time required to perform tasks that would otherwise be time-consuming for a certified FACS expert.

However, there are limitations to this study. The cross-sectional design may not allow us to find all causal inferences, and the findings may not be generalizable to all populations due to the specific demographic characteristics of the study sample. Additionally, while OpenFace is a powerful tool, it relies on clear and unobstructed video footage, and any deviations from this could impact the accuracy of the emotion analysis.

A further point concerns the applicability of FACS in infancy. While the original system was developed for adults, infant-adapted approaches such as Baby FACS have been developed to account for age-related differences in facial structure and expression, as described in a conference workshop [3]. Validation work for similar methods indicates that automated systems can capture infant expressions with reasonable accuracy, particularly for distinguishing positive from nonpositive states (AUC ≈ 0.81 in 4–8 month infants) [28]. Nevertheless, current datasets remain relatively small, and the developmental trajectory of facial musculature means that expression intensity and coordination may differ from adults. Our study contributes to this growing literature by applying automated analysis in a naturalistic feeding context, while recognizing that larger infant-specific datasets are needed to strengthen normative baselines.

Another limitation is that OpenFace struggles to separate AU10 (upper lip raiser) from AU12 (lip corner puller) due to overlapping muscle movements because of their overlapping muscle movements [4]. To address this, we combined AU9 + AU10 and AU6 + AU12 as markers for disgust and happiness, respectively. We thought that Including AU9 and AU6 would improve scoring accuracy since they are more reliably detected.

In the study, food selectivity was assessed solely based on the mother’s report, and no standardized questionnaire was used. It was observed that the reported foods included only certain specific items and did not comprehensively cover entire food groups. Also, our sample consisted of healthy infants aged 6–11 months. Therefore, findings may not generalize to malnourished or underserved populations with pediatric non-organic feeding disorders, where feeding responses may differ. Another important point is that our analysis was based on five predefined food groups rather than on individual taste qualities (like sweet, bitter, salty, sour, umami). Future research incorporating a broader range of taste exposures would be valuable to further investigate sensory-specific emotional responses.

## 5. Conclusions

Our findings indicate that infant emotional responses during complementary feeding are influenced by various mother–infant pair factors including maternal education level, BMI, employment status, timing of initial complementary feeding, and how food is prepared often to a greater extent than the type of food itself. This underscores the multifaceted nature of infant emotional development, where broader social and familial dynamics play a crucial role. Consequently, interventions aimed at addressing feeding difficulties should encompass these wider considerations.

While we did not measure longitudinal nutritional outcomes, infant emotions and parental responses likely influence dietary diversity and adequacy. Ultimately, good nutrition remains the core objective, and these emotional exchanges may play a pivotal role in shaping food exposure and long-term nutritional outcomes. Future studies could benefit from longitudinal designs to capture how these emotional reactions develop over time and to clarify how early feeding experiences affect later emotional and developmental trajectories.

Simultaneous observation of caregivers’ facial expressions with similar AI-powered methods during mealtimes may also offer valuable insights. Additionally, expanding studies to include more diverse populations would strengthen the generalizability of findings. Ongoing improvements in AI-driven tools like OpenFace could further refine their reliability and applicability in broader, less controlled environments, such as in-home feeding situations.

By deepening our understanding of how individual, social, and environmental elements interact with emotional responses, professionals can foster healthier feeding experiences and ultimately support more favorable developmental outcomes for infants.

## Figures and Tables

**Figure 1 nutrients-17-03182-f001:**
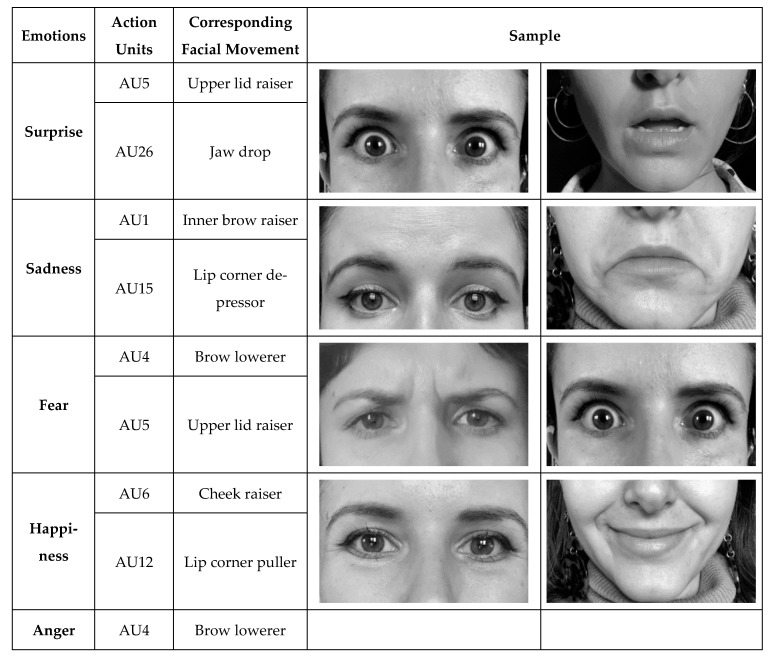
Emotions and Their Relevant Action Units, Muscle Groups and Facial Movement. Samples from an adult woman of corresponding facial movements and action units for six core emotions used in our study. Images used with permission.

**Figure 2 nutrients-17-03182-f002:**
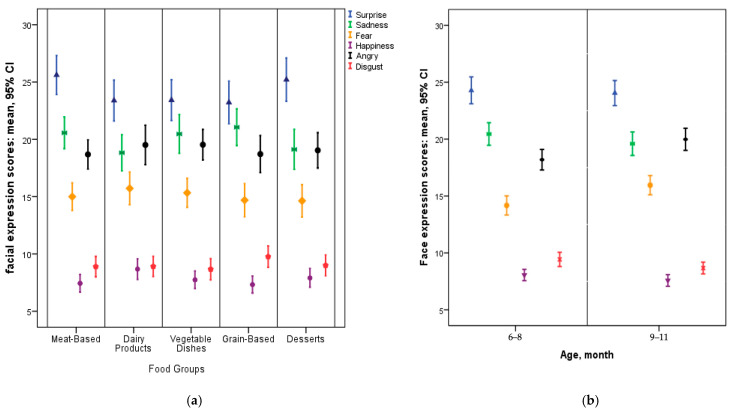
Mean Infant Emotional Expression Scores by Food and Age Group. Mean Infant Emotional Expression Scores (with 95% CI) by Complementary Food Groups (**a**), Mean Infant Emotional Expression Scores (95% CI) by Age Group (6–8 vs. 9–11 months) (**b**).

**Figure 3 nutrients-17-03182-f003:**
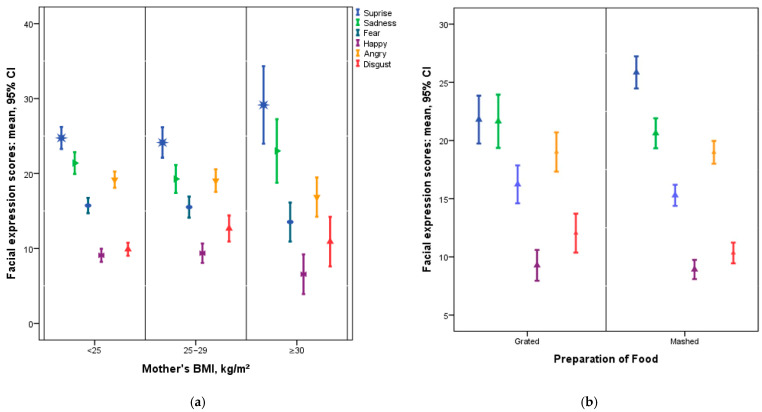
Mean Infant Emotional Expression Scores by Mother’s BMI and Food Preparation Style. Mean Infant Emotional Expression Scores (with 95% CI) by Mother’s BMI (kg/m^2^) (**a**), Mean Infant Emotional Expression Scores (95% CI) by Food Preparation Style (Grated vs. Mashed) (**b**).

**Table 1 nutrients-17-03182-t001:** General characteristics of study population, *n* = 117.

Mother’s Characteristics		Infant’s Characteristics	
Age, year	31.0 ± 4.2	Birth Order, %	
<30	36.8	First child	74.4
30–34	46.2	≥2nd child	25.6
≥35	17.1	Enrolled infant age, month	8.8 ± 2.0
Education status, %		6–8	51.3
≤High School	32.5	9–11	48.7
≥University	67.5	Gestational duration, week	38.7 ± 0.9
Working status, %		Birth weight, gram	3243 ± 403
Housewife	67.5	Sex	
Having a job	32.5	Male	48.7
BMI, kg/m^2^	24.5 ± 3.6	Female	51.3
<25	65.8	Initiation of Comp. Feeding	5.7 ± 0.5
25–29	28.2	≤4 months	1.7
>30	6.0	5 months	43.6
Has assistance for baby care, %	33.3	6 months	54.7
		Still Breastfed, %	
Food selectivity, %		Absence	32.5
Picky eater mother	10.3	Presence	67.5
Picky eater father	15.4	Weight-for-Age Z score	0.32 ± 0.88
At least one picky eater parent	23.1	<(−1)	6.0
Picky eater infant	28.2	(−1)–(+1)	71.8
		>(+1)	22.2
		Length-for-Age Z score	0.47 ± 1.22
		<(−1)	12.8
		(−1)–(+1)	56.4
		>(+1)	30.8
		Preparation of food, %	
		By Grating	24.8
		By Mashing	75.2

Values are given as percentage, mean ± standard deviation.

**Table 2 nutrients-17-03182-t002:** Facial Expression Intensities During Meat Consumption, *n* = 117.

Given Food Type	Preference	*n*	Surprise Mean [95% Wald CI]	Sadness Mean [95% Wald CI]	Fear Mean [95% Wald CI]	Happiness Mean [95% Wald CI]	Angry Mean [95% Wald CI]	Disgust Mean [95% Wald CI]
Minced meat	Dislike	14	26.2 [19.7–32.8]	21.5 [18.3–24.7]	15.1 [10.7–19.5]	6.5 [4.3–8.7]	19.3 [15.5–23.1]	7.7 [5.1–10.2]
	Like	30	26.4 [23.4–29.4]	20.9 [18.3–23.6]	13.8 [11.3–16.3]	7.3 [5.5–9.0]	16.7 [14–19.4]	9.4 [7.3–11.5]
Meatball	Dislike	10	29.7 [25.5–33.8]	22.3 [16.8–27.8]	11.9 [8.8–14.9]	7.1 [4.5–9.6]	17.5 [13.7–21.3]	9.4 [7.5–11.4]
	Like	18	26.1 [21.5–30.7]	20.5 [16.7–24.4]	16 [12.9–19.0]	7.6 [5.8–9.4]	18.9 [15.2–22.7]	7.8 [5.7–9.9]
*p*-value			0.738	0.924	0.403	0.904	0.594	0.551

Mean [95% CI]; one-way ANOVA.

**Table 3 nutrients-17-03182-t003:** Percentile distribution for emotional intensity measurements during feeding, *n* = 117.

	Mean	Std. Deviation	Percentiles		
			25	50	75
Surprise	24.2	9.4	17.8	24.1	30.5
Sadness	20.0	8.3	14.3	19.9	25.6
Angry	19.1	7.8	13.7	19.0	24.4
Fear	15.1	7.0	10.2	15.0	19.8
Happiness	7.8	4.2	4.8	7.7	10.6
Disgust	9.0	4.7	5.7	8.9	12.2

**Table 4 nutrients-17-03182-t004:** Emotional Responses to Different Food Groups During Complementary Feeding: Associations with Sociodemographic and Nutritional Characteristics in Infants Aged 6–11 Months Using Generalized Estimating Equations, *n* = 117 *.

	Surprise	Sadness	Fear	Happiness	Angry	Disgust
**Within-Subject**						
Food Group						
Meat-Based	25.5 [23.7–27.3]	21.0 [19.3–22.7]	15.3 [13.9–16.7]	6.7 [5.8–7.6]	18.9 [17.2–20.7]	8.9 [8.0–9.8]
Dairy Products	23.3 [21.4–25.2]	19.2 [17.2–21.3]	16.1 [14.4–17.8]	7.9 [7.0–8.8]	19.9 [17.8–21.9]	8.8 [7.7–9.9]
Vegetable Dishes	23.2 [21.2–25.2]	20.8 [18.8–22.8]	15.7 [14.3–17.2]	7.0 [6.0–7.9]	19.9 [18.1–21.7]	8.6 [7.6–9.6]
Grain-Based	23.1 [21.0–25.2]	21.5 [19.7–23.3]	15.1 [13.5–16.7]	6.6 [5.7–7.4]	19.1 [17.1–21.0]	9.7 [8.6–10.8]
Desserts	25.1 [22.8–27.5]	19.5 [17.4–21.6]	14.9 [13.3–16.6]	7.2 [6.2–8.1]	19.3 [17.3–21.3]	9.0 [7.9–10.0]
*p*	0.100	0.147	0.561	0.090	0.769	0.507
**Subject Variables**						
Mother’s Age, years						
<30	24.0 [22.3–25.7]	20.2 [18.2–22.1]	15.6 [13.8–17.4]	7.3 [6.3–8.3]	20.0 [18.0–22.0]	9.1 [8.1–10.0]
30–34	24.7 [23.1–26.2]	19.9 [18.0–21.8]	14.9 [13.3–16.5]	7.2 [6.3–8.1]	18.6 [16.9–20.4]	9.4 [8.4–10.4]
≥35	23.5 [21.2–25.7]	21.2 [19.3–23.0]	15.7 [13.6–17.9]	6.7 [5.4–7.9]	19.6 [17.1–22.2]	8.6 [7.5–9.6]
*p*	0.638	0.532	0.742	0.701	0.449	0.419
Education status						
<University	24.2 [22.4–26.1]	20.4 [18.5–22.4]	15.8 [14.1–17.6]	6.3 [5.4–7.3]	19.9 [17.7–22.0]	8.7 [7.6–9.8]
≥University	23.9 [22.4–25.3]	20.4 [18.8–22.0]	15.0 [13.6–16.5]	7.8 [7.0–8.6]	19.0 [17.4–20.6]	9.3 [8.6–10.0]
*p*	0.719	0.953	0.428	**0.008**	0.421	0.289
Working status						
Non-working	24.2 [22.6–25.8]	20.3 [18.8–21.8]	14.5 [12.9–16.0]	7.7 [7.0–8.5]	18.1 [16.3–19.9]	9.6 [8.8–10.4]
Working	23.9 [22.1–25.7]	20.5 [18.5–22.5]	16.4 [14.7–18.1]	6.4 [5.4–7.4]	20.7 [18.7–22.7]	8.4 [7.5–9.3]
*p*	0.732	0.832	0.086	**0.019**	**0.026**	**0.025**
Mother’s BMI,						
<25 km/m^2^	23.0 ^a^ [21.4–24.6]	20.7 [19.1–22.2]	16.1 [14.6–17.7]	7.4 [6.6–8.3]	19.9 [18.2–21.7]	8.3 ^a^ [7.4–9.1]
25–29 km/m^2^	22.8 ^a^ [20.9–24.8]	19.1 [17.2–21.1]	15.9 [13.9–17.9]	7.7 [6.7–8.8]	20.1 [17.8–22.4]	9.6 ^b^ [8.7–10.5]
≥30 km/m^2^	26.3 ^b^ [24.2–28.4]	21.4 [18.9–24.0]	14.3 [12.4–16.1]	6.0 [4.7–7.3]	18.2 [16.2–20.3]	9.2 ^ab^ [7.7–10.6]
*p*	**0.007**	0.208	0.206	0.074	0.247	**0.050**
Birth order						
First	23.6 [22.2–25.0]	20.3 [18.8–21.8]	15.3 [13.9–16.6]	7.3 [6.4–8.1]	19.4 [17.9–20.9]	9.4 [8.6–10.2]
Non-first	24.5 [22.7–26.3]	20.5 [18.5–22.5]	15.6 [13.8–17.4]	6.9 [5.9–7.8]	19.4 [17.2–21.6]	8.7 [7.7–9.6]
*p*	0.386	0.906	0.725	0.455	0.988	0.191
Infant Age						
6–8 mo	24.2 [22.7–25.8]	20.9 [19.2–22.5]	14.5 [13.0–16.0]	7.4 [6.5–8.3]	18.4 [16.6–20.2]	9.3 [8.3–10.3]
9–11 mo	23.9 [22.3–25.5]	20 [18.3–21.7]	16.3 [14.7–18.0]	6.7 [5.9–7.6]	20.4 [18.6–22.2]	8.7 [7.9–9.5]
*p*	0.696	0.329	0.069	0.214	**0.050**	0.277
Sex						
Male	23.7 [22.1–25.3]	20.2 [18.4–22.1]	15.5 [13.9–17.1]	7 [6.1–7.8]	19.9 [18.1–21.7]	8.9 [8–9.9]
Female	24.3 [22.8–25.9]	20.6 [19.2–22]	15.4 [13.9–16.8]	7.2 [6.3–8.0]	18.9 [17.2–20.7]	9.1 [8.3–9.8]
*p*	**0.002**	0.685	0.896	0.677	0.312	0.816
Initiation of Comp. feeding						
<6 mo	23.6 [21.5–25.6]	20.1 [18.0–22.2]	16.1 [14.1–18]	6.5 [5.5–7.6]	20.7 [18.4–22.9]	8.8 [7.8–9.8]
6 mo	24.5 [23.2–25.8]	20.8 [19.3–22.2]	14.8 [13.5–16.1]	7.6 [6.8–8.4]	18.2 [16.7–19.7]	9.2 [8.4–10.0]
*p*	0.410	0.540	0.256	0.069	**0.034**	0.538
Still Breastfed						
Absence	23.8 [22.2–25.5]	20.3 [18.8–21.9]	15.1 [13.6–16.6]	7.5 [6.5–8.4]	19.3 [17.6–21.1]	8.9 [8–9.8]
Presence	24.3 [22.7–25.8]	20.5 [18.7–22.3]	15.7 [14.1–17.4]	6.7 [5.8–7.5]	19.5 [17.6–21.4]	9.1 [8.2–10]
*p*	0.632	0.879	0.566	0.143	0.852	0.734
Preparation of Food						
Grated	22.6 [20.7–24.4]	20.7 [18.7–22.8]	15.9 [13.9–17.9]	7.2 [6.1–8.2]	19.6 [17.2–22.0]	9.5 [8.6–10.5]
Mashed	25.5 [24.2–26.8]	20.1 [18.7–21.5]	15.0 [13.7–16.2]	7.0 [6.2–7.7]	19.2 [17.9–20.5]	8.5 [7.7–9.3]
*p*	**0.002**	0.550	0.434	0.739	0.735	0.064
WAZ						
≤1 z score	24.4 [23.0–25.7]	20.6 [19.0–22.2]	15.4 [13.9–16.8]	6.8 [5.9–7.7]	19.7 [17.9–21.5]	8.9 [8.1–9.8]
>1 z score	23.7 [21.7–25.7]	20.2 [18.2–22.3]	15.5 [13.6–17.4]	7.3 [6.3–8.4]	19.1 [17.0–21.3]	9.1 [8.1–10.1]
*p*	0.569	0.765	0.916	0.390	0.649	0.818

* Values given as estimated mean (95% Wald Confidence Interval); ^ab^ values with different letters in the same column were found to be statistically different, *p* < 0.05. Bold *p* values indicate statistical significance.

**Table 5 nutrients-17-03182-t005:** Emotional Responses to Different Food Groups During Complementary Feeding in Infants Aged 6–11 Months: Associations with Parental and Infant Food Selectivity Using Generalized Estimating Equations, *n* = 117 *.

	Surprise	Sadness	Fear	Happiness	Angry	Disgust
Food Group						
Meat-Based	25.7 [23.9–27.6]	20.8 [19.4–22.2]	14.8 [13.5–16]	7.1 [6.3–7.9]	18.9 [17.6–20.1]	9.0 [8.1–10]
Dairy Products	23.5 [21.6–25.4]	19 [17.4–20.6]	15.6 [14.2–16.9]	8.3 [7.4–9.2]	19.8 [18.1–21.5]	9.0 [8.1–9.9]
Vegetable Dishes	23.5 [21.6–25.5]	20.7 [18.8–22.5]	15.2 [13.8–16.6]	7.4 [6.5–8.2]	19.8 [18.3–21.4]	8.8 [7.8–9.7]
Grain-Based	23.3 [21.3–25.4]	21.3 [19.6–23]	14.5 [13–16]	7 [6.1–7.8]	18.9 [17.2–20.7]	9.9 [8.9–10.9]
Desserts	25.4 [23.3–27.4]	19.4 [17.5–21.2]	14.4 [12.9–15.8]	7.5 [6.7–8.4]	19.2 [17.6–20.8]	9.1 [8.2–10.1]
*p*	0.095	0.148	0.541	0.078	0.773	0.472
Infant Age, mo						
6–8	24.5 [23.0–26.0]	20.7 [19.3–22.1]	13.9 [12.6–15.3]	7.7 [6.9–8.4]	18.4 [17–19.9]	9.5 [8.7–10.3]
9–11	24.1 [22.7–25.5]	19.8 [18.5–21.1]	15.8 [14.4–17.2]	7.3 [6.5–8]	20.2 [18.8–21.7]	8.8 [8.1–9.5]
*p*	0.692	0.320	**0.041**	0.424	0.061	0.125
Infants’ selective eating						
Not Present	24 [22.7–25.3]	20.1 [19–21.1]	15.3 [14.2–16.4]	7.8 [7.2–8.3]	19.3 [18.1–20.6]	9.2 [8.6–9.8]
Present	24.6 [22.9–26.3]	20.4 [18.7–22.1]	14.5 [12.8–16.1]	7.1 [6.1–8.2]	19.4 [17.7–21]	9.1 [8.2–10.0]
*p*	0.543	0.786	0.421	0.275	0.962	0.800
Parental selective eating						
Not Present	24.3 [23.3–25.3]	20 [18.9–21]	14.9 [13.8–16]	7.8 [7.1–8.5]	18.9 [17.8–20]	8.9 [8.3–9.5]
Present	24.3 [22.2–26.4]	20.5 [18.8–22.2]	14.9 [13.2–16.5]	7.1 [6.3–7.9]	19.8 [17.9–21.6]	9.4 [8.5–10.4]
	0.953	0.582	0.985	0.121	0.428	0.315

* Values given as estimated mean (95% Wald Confidence Interval). Bold *p* values indicate statistical significance.

## Data Availability

The datasets presented in this article are not readily available due to the inclusion of identifiable images of children in the dataset, we did not obtain consent from the families to share this data with third parties. Consent was granted solely for the use of the data within the scope of this study.

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
