# Peer review of "AI-Based Facial Emotion Analysis in Infants During Complimentary Feeding: A Descriptive Study of Maternal and Infant Influences"

_nutrients, 2025, doi:10.3390/nu17193182_

Round 1

Reviewer 1 Report

Comments and Suggestions for Authors

General comments: 

The authors' research is not new.  Similar research was conducted at the Monell Chemical Senses Center approximately 40 years ago.  That work was not cited by the authors.  At least the authors cited the important early work by Ekman et al.   The results from these researchers are similar to that presented in the current manuscript.  The only differences are the tools to assess the facial and emotional outcomes.

L34: Introduction: (this section is well written)

L55: Figure 1 clearly depicts expressions by adult women.  Similar photos of infants and children (with appropriate masking) would add considerable value to the manuscript.

L66: the citation of Romano et al (10) is good; one challenge is that the typical NOFED presentations differ among children (< 6 yo) who are malnourished and underserved.  Authors must address this point.

L93: good comment related to infants 6-11 mos; this deserved greater attention; remember, human milk is "sweet" to most infants despite the low free glucose content.

L100: interesting Ho advanced by the authors

L110: Materials & Methods:  the narrative in this section is reasonable.

L138: good to see IRB approval; what about pt consent, e.g., Declaration of Helsinki consent for clinical studies?

L187: Results

L188: Table 1 is classic study demographics; note, typically in infants, the HAZ is really Length-for-Age Z-score; supprised no infants < 6 mos despite ~44% of infants received comp feeds at 5 mos.

L192: other than Z scores, what additional tools were used to assess nutritional status of infants and their mothers? 

L227: this reviewer is surprised that none of the ensuing tables included responses to classic sensory/organoleptic sensations, e.g., sweet, bitter, salty, sour, umami... in addition to texture, visual, smell (at least for adults)

L193: how did the authors define "picky" eating patterns?

L287: Discussion: 

L289: how is AI applied to the data?  the presented data a classic for this kind of senory study; if AI is used, how did the authors validate the algorithm(s) for applications suggested in this study?

L317:  Interesting comments regarding anger and working mothers.  In the US, there is a law that encourages lactating women to continue feeds during work hours and at the place of employment.  Employers actually are eligible for funding from the Federal government for these kinds of support. (see Providing Urgent Maternal Protections for Nursing Mothers Act (aka PUMP Act of 2022)

L437: the strengths/limitations section is reasonable

L437: Conclusions

L438: delete the first sentence; the remaining narrative in this section is reasonable

Author Response

Dear reviewer and editor,

Thank you for your valuable contributions and thoughtful suggestions regarding our manuscript. We have adjusted the formatting of Tables 4 and 5 to improve fit, specifically by modifying the placement of the p-values. We also added more references to comply with peer reviewers’ suggestions. Our detailed responses are provided below.

Reviewer 1:

Comments-1: The authors' research is not new.  Similar research was conducted at the Monell Chemical Senses Center approximately 40 years ago.  That work was not cited by the authors.  At least the authors cited the important early work by Ekman et al.   The results from these researchers are similar to that presented in the current manuscript.  The only differences are the tools to assess the facial and emotional outcomes.

Response-1: Thank you for this important comment. We agree that foundational work on infant facial responses to taste was conducted several decades ago. To reflect this, we cited Rosenstein and Oster’s later chapter “Rosenstein, D.; Oster, H. Differential Facial Responses to Four Basic Tastes in Newborns. In What the Face Reveals: Basic and Applied Studies of Spontaneous Expression Using the Facial Action Coding System (FACS); Oxford University Press, 2012” which builds directly on their earlier publication: “Rosenstein D, Oster H. Differential facial responses to four basic tastes in newborns. Child Dev. 1988;59(6):1555–1568”. We chose the 2012 reference at first because it explicitly connects this work to the Facial Action Coding System (FACS), which was central to our methodology. But to comply with your opinion, we also added the first work.

As you rightly point out, our study does not introduce an entirely new conceptual approach to investigating whether newborns produce distinct facial expressions in response to tastes, nor to developing methods for coding facial - emotion relations. Rather, the novelty of our work lies in applying an artificial intelligence tool, OpenFace, to efficiently assess infants’ facial expressions during feeding. Unlike traditional FACS coding, which requires trained experts and is time-intensive, OpenFace provides an accessible and scalable method that can be implemented by clinicians and researchers without specialized training. Our aim was to demonstrate the feasibility of this approach and to open a pathway for broader use of automated tools in studies of infant feeding behavior, while also exploring associations between emotional responses during feeding and other relevant factors.

Comments-2: L34: Introduction: (this section is well written)

L100: interesting Ho advanced by the authors

L110: Materials & Methods:  the narrative in this section is reasonable.

L317: Interesting comments regarding anger and working mothers.  In the US, there is a law that encourages lactating women to continue feeds during work hours and at the place of employment.  Employers actually are eligible for funding from the Federal government for these kinds of support. (see Providing Urgent Maternal Protections for Nursing Mothers Act (aka PUMP Act of 2022)

L437: the strengths/limitations section is reasonable

Response-2: Thank you for these comments and information

Comments-3: L55: Figure 1 clearly depicts expressions by adult women.  Similar photos of infants and children (with appropriate masking) would add considerable value to the manuscript.

Response-3: Thank you for this insightful comment. We acknowledge that examples featuring infants or children could further enrich the manuscript. However, reliably capturing and illustrating the relevant Action Units in children poses significant challenges. For this reason, we used pictures of an adult woman who is a certified Facial Action Coding System (FACS) expert, and all images were used with her consent. The purpose of including this figure was to clearly demonstrate the specific AUs analyzed and the emotions they represent, thereby providing readers with a consistent reference framework.

Comments-4: L66: the citation of Romano et al (10) is good; one challenge is that the typical NOFED presentations differ among children (< 6 yo) who are malnourished and underserved.  Authors must address this point.

Response-4: We added a limitation stating our findings may not generalize to malnourished or underserved populations with NOFED as follows in page 15: “Also, our sample consisted of healthy infants aged 6–11 months. Therefore, findings may not generalize to malnourished or underserved populations with pediatric non-organic feeding disorders, where feeding responses may differ.”

Comments-5: L93: good comment related to infants 6-11 mos; this deserved greater attention; remember, human milk is "sweet" to most infants despite the low free glucose content.

Response: We add this to discussion: “The higher lactose content of human milk compared to the milk of other mammals may influence the infant's taste preferences in complementary feeding [23].”

Comments-6: L138: good to see IRB approval; what about pt consent, e.g., Declaration of Helsinki consent for clinical studies?

Response-6: We clarified that informed consent was obtained from all parents in accordance with the Declaration of Helsinki in the last section of the study (page 16) as follows:

“The  Institutional Review Board Statement: The study was conducted in accordance with the Declaration of Helsinki, and ethical approval was granted by the BaÅŸkent University Ethics Committee (KA24/68-24/54, 28th February 2024).

Informed Consent Statement: Informed consent was obtained from all subjects involved in the study.”

Comments 7: L188: Table 1 is classic study demographics; note, typically in infants, the HAZ is really Length-for-Age Z-score; supprised no infants < 6 mos despite ~44% of infants received comp feeds at 5 mos.

Response-7:  We changed terminology in the text and tables. Also our study specifically included infants aged 6–11 months at the time of enrollment. Information regarding the initiation of complementary feeding was collected retrospectively from caregivers. Thus, while some infants had begun complementary feeding before 6 months, our analysis focused on the predefined study age range of 6–11 months.

We also added this important point you mentioned to discussion: “In our study, approximately half of the infants had started complementary feeding before the 6th month. This finding is consistent with the current literature in our country.

Comments-8 L192: other than Z scores, what additional tools were used to assess nutritional status of infants and their mothers?

Response-8: We calculated Weight-for-Age Z score, Length-for-Age Z score for infants and BMI for mothers. No additional tools were used.

Comments-9: L227: this reviewer is surprised that none of the ensuing tables included responses to classic sensory/organoleptic sensations, e.g., sweet, bitter, salty, sour, umami... in addition to texture, visual, smell (at least for adults).

Response-9: Thank you for mentioning this point. We acknowledged that our design was based on food groups, not isolated taste qualities, and flagged this as future research in page 15 as: “Another important point is that our analysis was based on five predefined food groups rather than on individual taste qualities (like sweet, bitter, salty, sour, umami). Future research incorporating a broader range of taste exposures would be valuable to further investigate sensory-specific emotional responses.”

Comments-10: L193: how did the authors define "picky" eating patterns?

Response-10: Thank you for this comment. We have now added the definition we used for picky eating to the manuscript from the relevant literature: “Although there is no universally accepted definition, we defined picky eaters as children who consume an insufficient variety of foods while rejecting substantial amounts of both familiar and unfamiliar foods.”

Comments-11 L289: how is AI applied to the data?  the presented data a classic for this kind of senory study; if AI is used, how did the authors validate the algorithm(s) for applications suggested in this study?

Response-11: We applied AI to the on the videos of infants using the OpenFace, an established open-source program for facial behavior analysis. The software detects and aligns faces, identifies facial landmarks, and uses its trained models to classify Facial Action Units (AUs) and quantify their intensity. OpenFace has been widely validated against human-coded Facial Action Coding System (FACS) annotations in multiple contexts, including psychological and behavioral studies. In our study, we relied on these rather than developing new ones and applied them consistently across all participants to ensure comparability of results.

Comments-12 L438: delete the first sentence; the remaining narrative in this section is reasonable

Response-12: Leftover sentence is deleted.

Reviewer 2 Report

Comments and Suggestions for Authors

A very innovative and interesting paper, yet one which raises some issues which could strengthen it:

  1. The description of FACS is good, but the example pictures are of adults. Discussion is needed as to whether FACS has been assessed or validated in infants (noting that the authors had to improvise their own modification). Are the physiological mechanisms and responses fully developed in infancy, or do they modify into adulthood on which the base FACS research is grounded?
  2. Might the study mothers' selection and preparation of the foods for the study be biased because the mother felt under scrutiny?  Might she have prepared a different, more 'proper' or careful in her eyes version of the food the child knew from the domestic setting?  This could cause a reaction if a 'familiar' food in fact was different to taste or composition.
  3. The tables of responses are clear, but the relative factors are difficult to assimilate.  Could visual or diagrammatic displays be considered?
  4. Satisfaction with food is of course important to ensure continued good nutrition patterns.  But are the effects of emotions known - for instance, is Surprise unwelcome, or welcome and a stimulant for hopes of further new experiences?
  5. Why did the study apparently not consider measuring duration of continued intake of each food?  Did type of emotion have any affect on this?
  6. It seems the loop needs closing between parental choice of foods, infants' reactions, parents' response to reactions, and subsequent feeding choices and patterns.
  7. Finally, the core objective is good nutrition.  Did the expressed emotions on food types, or parents' interpretations and reactions, have effects on overall nutrition?

Language and presentation were excellent, but some instructions to authors appear in text of headings of some tables and paragraphs.

Author Response

Dear reviewers and editor,

Thank you for your valuable contributions and thoughtful suggestions regarding our manuscript. We have adjusted the formatting of Tables 4 and 5 to improve fit, specifically by modifying the placement of the p-values. We also added more references to comply with peer reviewers’ suggestions. Our detailed responses are provided below and the new version of article added as a new file named: "Feeding-nutrients rev1"

Reviewer 2:

A very innovative and interesting paper, yet one which raises some issues which could strengthen it:

Comments-1: The description of FACS is good, but the example pictures are of adults. Discussion is needed as to whether FACS has been assessed or validated in infants (noting that the authors had to improvise their own modification). Are the physiological mechanisms and responses fully developed in infancy, or do they modify into adulthood on which the base FACS research is grounded?

Response-1: Reliably capturing and illustrating the relevant Action Units in children poses significant challenges. For this reason, we used pictures of an adult woman who is a certified Facial Action Coding System (FACS) expert, and all images were used with her consent. The purpose of including this figure was to clearly demonstrate the specific AUs analyzed and the emotions they represent, thereby providing readers with a consistent reference framework.

We added the relevant paragraph in discussion with new references to mention the points you provided as follows: “A further point concerns the applicability of FACS in infancy. While the original system was developed for adults, infant adapted approaches such as Baby FACS have been developed to account for age-related differences in facial structure and expression, as described in a conference workshop [23]. Validation work for similar methods indicates that automated systems can capture infant expressions with reasonable accuracy, particularly for distinguishing positive from nonpositive states (AUC ≈ 0.81 in 4–8 month infants) [24]. Nevertheless, current datasets remain relatively small, and the develop-mental trajectory of facial musculature means that expression intensity and coordination may differ from adults. Our study contributes to this growing literature by applying automated analysis in a naturalistic feeding context, while recognizing that larger infant specific datasets are needed to strengthen normative baselines”

Comments-2: Might the study mothers' selection and preparation of the foods for the study be biased because the mother felt under scrutiny?  Might she have prepared a different, more 'proper' or careful in her eyes version of the food the child knew from the domestic setting?  This could cause a reaction if a 'familiar' food in fact was different to taste or composition.

Response-2: Thank you for this thoughtful comment. Our aim was to capture infants’ emotional responses as naturally as possible. Therefore, mothers prepared the selected foods and recorded the feeding sessions in their home environment. This approach was intended to minimize potential bias by allowing families to use their usual preparation practices. This situation stated in the study protocol: “The feeding sessions were recorded by the mothers and provided at subsequent appointments.”

Comments-3: The tables of responses are clear, but the relative factors are difficult to assimilate.  Could visual or diagrammatic displays be considered?

Respons-3: Thank you for this helpful suggestion. We initially considered adding visual or diagrammatic displays; however, we concluded that tables provide the most comprehensive format, as they allow us to present all assessments in detail. Adding figures in addition to the tables would considerably increase the manuscript length, while replacing tables with figures would reduce the amount of data we could present. For these reasons, we felt that the table format was the most suitable for conveying the results.

Comments-4: Satisfaction with food is of course important to ensure continued good nutrition patterns.  But are the effects of emotions known - for instance, is Surprise unwelcome, or welcome and a stimulant for hopes of further new experiences?

Response-4: Thank you for raising this important point. The role of emotions in shaping feeding experiences is not yet fully understood. Certain emotions, such as Anger, Disgust, and Happiness, can more readily be associated with either food aversion or acceptance. In contrast, Surprise is more ambiguous, as it may signal positive curiosity or negative uncertainty. To our knowledge, no clear link between Surprise and feeding outcomes has been established in the literature. In our study, we therefore treated Surprise as a neutral category, not directly associated with acceptance or rejection. (For example: “The slightly elevated Surprise in female infants compared to males could reflect a potential gender based disparity in expressiveness or sensitivity to taste stimuli” or “Mothers with higher BMI might offer a broader array of foods or employ distinct feeding techniques that provoke greater surprise, possibly due to novel flavors or unexpected textures”)

Comments-5: Why did the study apparently not consider measuring duration of continued intake of each food?  Did type of emotion have any affect on this?

Response-5: As described in Section 2.2 (Capturing and processing the video), each of the five food categories was offered in sequence, with sufficient time for the child to chew and swallow before proceeding. Our analysis focused on infants’ facial reactions within a single cycle of each food group. Because chewing and swallowing times vary across both children and food types, we applied time-filtered averaging on the usable video frames to obtain a representative value for each Action Unit intensity. Since our objective was not to measure the duration of consumption or to fully feed the child with each item, but rather to capture immediate emotional reactions, the length of continued intake was not considered a relevant outcome in this study. However, we appreciate this suggestion and recognize that examining intake duration in relation to emotional responses could be an important area for our future research.

Comments-6: It seems the loop needs closing between parental choice of foods, infants' reactions, parents' response to reactions, and subsequent feeding choices and patterns.

Response-6: Thank you for that point. We added a paragraph in section 4.2 regarding to state this: “Importantly, feeding is not a one directional process but a dynamic feedback loop. Parents make choices about what foods to offer, infants respond emotionally, and these reactions shape parents’ subsequent decisions about feeding. For instance, if a child consistently shows anger or disgust toward a certain food, a parent may avoid offering it in the future, while repeated expressions of happiness may reinforce continued provision of that food. Over time, these reciprocal interactions accumulate into recognizable feeding patterns and dietary preferences. Highlighting this cyclical nature of feeding underscores that infant emotions are not only outcomes of parental decisions but also active signals that guide future parental choices.”

Comments-7: Finally, the core objective is good nutrition.  Did the expressed emotions on food types, or parents' interpretations and reactions, have effects on overall nutrition?

Response-7: Thank you for emphasizing this. In our dataset, expressed emotions did not vary by food type, and they did not correlate with concurrent nutritional status (WAZ). We also observed no associations between infant or parental selective eating and infants’ emotional responses. Thus, within this cross-sectional sample, we found no evidence that expressed emotions by food type or parental selectivity were linked to overall nutrition at the time of measurement. We did not collect dietary intake/diversity or longitudinal outcomes, and we did not code feeder reactions; therefore, we cannot infer causal effects of parental interpretations or reactions on nutrition. We now clarify the conclusion paragraph to emphasize this question for further studies: “While we did not measure longitudinal nutritional outcomes, infant emotions and parental responses likely influence dietary diversity and adequacy. Ultimately, good nutrition remains the core objective, and these emotional exchanges may play a pivotal role in shaping food exposure and long term nutritional outcomes. Future studies could benefit from longitudinal designs to capture how these emotional reactions develop over time and to clarify how early feeding experiences affect later emotional and developmental trajectories.”

Comments-8: Language and presentation were excellent, but some instructions to authors appear in text of headings of some tables and paragraphs.

Response-8: Thank you for this comment. Overlooked instructions deleted.

Round 2

Reviewer 2 Report

Comments and Suggestions for Authors

The response to reviewer comments has been rapid and generally effective.

Regarding Comment 1, some reference to FACS and Baby FACS should be made when FACS is first introduced.  Leaving this until the Discussion stage in toto is too late.  So the helpful material currently added needs to be split between an outline comment in Methods, and more detail in Discussion.

Regarding Comment 2, the words 'at home' might helpfully be added in line 142 about mothers self-recording.  However, the general construct between that section, and the beginning of the next section on data capture, might be revisited as the mothers' recording is part of data capture.  Also, given that this was a participant at home action, some brief mention should be given to explain how this specification was explained to the mothers, and comment made about how well this requirement was met, or whether any videos had to be repeated because of errors in positioning or lighting making specific results difficult to interpret.

Regarding Response 3, the views of the editor are sought about the balance between manuscript length and the potential added clarity possible with addition of visual displays.

For all other comments, the resultant Responses and actions are welcomed and accepted.

Author Response

Dear reviewers and editor,

Thank you for your valuable contributions and thoughtful suggestions regarding our manuscript. We also added more references to comply with peer reviewers’ suggestions. Our detailed responses are provided below.

Reviewer 2:

Comments-1 (prior): The description of FACS is good, but the example pictures are of adults. Discussion is needed as to whether FACS has been assessed or validated in infants (noting that the authors had to improvise their own modification). Are the physiological mechanisms and responses fully developed in infancy, or do they modify into adulthood on which the base FACS research is grounded?

Response-1 (prior): Reliably capturing and illustrating the relevant Action Units in children poses significant challenges. For this reason, we used pictures of an adult woman who is a certified Facial Action Coding System (FACS) expert, and all images were used with her consent. The purpose of including this figure was to clearly demonstrate the specific AUs analyzed and the emotions they represent, thereby providing readers with a consistent reference framework.

We added the relevant paragraph in discussion with new references to mention the points you provided as follows: “A further point concerns the applicability of FACS in infancy. While the original system was developed for adults, infant adapted approaches such as Baby FACS have been developed to account for age-related differences in facial structure and expression, as described in a conference workshop [23]. Validation work for similar methods indicates that automated systems can capture infant expressions with reasonable accuracy, particularly for distinguishing positive from nonpositive states (AUC ≈ 0.81 in 4–8 month infants) [24]. Nevertheless, current datasets remain relatively small, and the develop-mental trajectory of facial musculature means that expression intensity and coordination may differ from adults. Our study contributes to this growing literature by applying automated analysis in a naturalistic feeding context, while recognizing that larger infant specific datasets are needed to strengthen normative baselines”

Regarding Comment 1: some reference to FACS and Baby FACS should be made when FACS is first introduced.  Leaving this until the Discussion stage in toto is too late.  So the helpful material currently added needs to be split between an outline comment in Methods, and more detail in Discussion.

Response to new Comment-1: We thank the reviewer for this valuable suggestion. In line with the comment, we revised the Introduction to briefly note the existence of Baby FACS when introducing FACS, while retaining the more detailed discussion of its applicability, validation, and limitations in the Discussion section. This ensures readers are aware from the outset that infant adaptations exist. Here is the added part in introduction:

“While originally developed for adults, infant-adapted approaches such as Baby FACS have since been introduced to account for age-related differences in facial structure and expression [3]. Nevertheless, the fundamental AU framework remains highly relevant and provides the basis for automated facial analysis”

Comments-2 (prior): Might the study mothers' selection and preparation of the foods for the study be biased because the mother felt under scrutiny?  Might she have prepared a different, more 'proper' or careful in her eyes version of the food the child knew from the domestic setting?  This could cause a reaction if a 'familiar' food in fact was different to taste or composition.

Response-2 (prior): Thank you for this thoughtful comment. Our aim was to capture infants’ emotional responses as naturally as possible. Therefore, mothers prepared the selected foods and recorded the feeding sessions in their home environment. This approach was intended to minimize potential bias by allowing families to use their usual preparation practices. This situation stated in the study protocol: “The feeding sessions were recorded by the mothers and provided at subsequent appointments.”

Regarding Comment 2: the words 'at home' might helpfully be added in line 142 about mothers self-recording.  However, the general construct between that section, and the beginning of the next section on data capture, might be revisited as the mothers' recording is part of data capture.  Also, given that this was a participant at home action, some brief mention should be given to explain how this specification was explained to the mothers, and comment made about how well this requirement was met, or whether any videos had to be repeated because of errors in positioning or lighting making specific results difficult to interpret.

Response to new Comment-2: Thank you for the suggestion. We have revised 2.1 Study Protocol to state explicitly that mothers self-recorded sessions at home and to summarize the standardized guidance they received: “The feeding sessions were self-recorded by the mothers at home and provided at subsequent appointments. Mothers were given standardized instructions on camera placement, lighting, and distance (approximately one meter from the infant) to maximize video quality. If a submitted video did not meet these criteria (e.g., major face occlusion or poor illumination), a repeat recording was requested.”

Comments 3 (prior): The tables of responses are clear, but the relative factors are difficult to assimilate.  Could visual or diagrammatic displays be considered?

Response3 (prior): Thank you for this helpful suggestion. We initially considered adding visual or diagrammatic displays; however, we concluded that tables provide the most comprehensive format, as they allow us to present all assessments in detail. Adding figures in addition to the tables would considerably increase the manuscript length, while replacing tables with figures would reduce the amount of data we could present. For these reasons, we felt that the table format was the most suitable for conveying the results.

Regarding Response 3: the views of the editor are sought about the balance between manuscript length and the potential added clarity possible with addition of visual displays.

Response to new Comment 3:

We appreciate the reviewer’s further note on this point. Although our initial view was that tables provided the most comprehensive presentation, we have now added an additional figure-3 depicting mother’s BMI and food preparation style in response to the reviewer’s suggestion, in order to enhance clarity.